# Fine-Scaled Selection of Resting and Hunting Habitat by Leopard Cats (*Prionailurus bengalensis*) in a Rural Human-Dominated Landscape in Taiwan

**DOI:** 10.3390/ani13020234

**Published:** 2023-01-08

**Authors:** Esther van der Meer, Hans Dullemont, Ching-Hao Wang, Jun-Wei Zhang, Jun-Liang Lin, Kurtis Jai-Chyi Pei, Yu-Ching Lai

**Affiliations:** 1Institute of Wildlife Conservation, College of Veterinary Medicine, National Pingtung University of Science and Technology, Pingtung 912301, Taiwan; 2Department of Landscape Architecture and Environmental Design, Huafan University, New Taipei City 223, Taiwan

**Keywords:** leopard cat, refuge habitat, habitat selection, human wildlife co-existence, anthropogenic mortality, domestic dogs

## Abstract

**Simple Summary:**

Presently, Taiwan’s endangered leopard cat mostly occurs in unprotected, rural, human-dominated landscapes. To survive within these landscapes, the species needs suitable habitat for nighttime hunting but also safe refuge for resting during daytime hours when human activity, and herewith human-related threat, peaks. Although important for the species’ conservation, little is known about the characteristics of the leopard cat’s resting habitat. In this study, we tracked seven VHF-collared leopard cats. Every day we determined where these leopard cats rested, and every ten days we followed a leopard cat throughout the night and recorded its locations at 30 min intervals. We assessed land use at nighttime locations and land use and fine-scaled vegetation characteristics at resting sites and determined whether leopard cats selected certain habitats or simply used them according to availability. The leopard cats in our study seemed to use a proactive strategy to avoid humans by selecting natural habitats for hunting and resting and avoiding areas with high levels of human activity. Resting sites were preferably situated in natural habitats with little visibility (<2 m), shrubs, reed and stones, away from orchards, buildings and roads. This information will assist in identifying and conserving suitable resting habitats for leopard cats.

**Abstract:**

Wildlife is increasingly forced to live in close proximity to humans, resulting in human-wildlife conflict and anthropogenic mortality. Carnivores persisting in human-dominated landscapes respond to anthropogenic threats through fine-scaled spatial and temporal behavioral adjustments. Although crucial for conservation, quantitative information on these adjustments is scarce. Taiwan’s endangered leopard cat occurs in rural human-dominated landscapes with a high anthropogenic mortality risk. To survive, the nocturnal leopard cat needs suitable habitats for foraging and safe refuge for resting during daytime hours when human activity peaks. In this study, we tracked seven VHF-collared leopard cats. To determine habitat selection patterns, we compared land use at nighttime locations and daytime resting sites with random points and fine-scaled vegetation characteristics at daytime resting sites with random points. Leopard cats selected natural habitats for nighttime hunting and avoided manmade and, to a lesser extent, agricultural habitats or used them according to availability. For daytime resting, leopard cats selected natural habitats and, to a lesser extent semi-natural habitats, such as unused land and abandoned orchards. Resting sites were preferentially situated in natural habitats, with little visibility (<2 m), shrubs, reed and stones, away from areas with high levels of human activity. This suggests leopard cats use a proactive strategy to avoid human encounters, which was supported by the reduced temporal overlap with humans and domestic dogs on agricultural land. Resting sites were placed ca. 1 km apart, 12.9 ± 0.3 m (mean ± SE) from the patch’s edges, in patches with a size of 1.21 ± 0.04 ha (mean ± SE). Our results will assist in identifying and preserving suitable resting habitats to support leopard cat conservation.

## 1. Introduction

An increase in the world’s human population has resulted in the degradation, loss and fragmentation of habitats available to wildlife [1,2]. Small patches of remaining wildlife habitats are often embedded within a human-modified matrix [3], forcing wildlife species to live in close proximity to humans [4,5]. For mammalian carnivores especially, this has resulted in an increase in human–wildlife conflict and anthropogenic mortality, which is considered one of the most pressing global carnivore conservation issues today [2,4,5].

In Taiwan, human population growth and speedy industrial development has had a marked impact on wildlife [6]. With ca. 500 individuals remaining [7], the leopard cat *(Prionailurus bengalensis)* (2–4 kg) is now Taiwan’s only indigenous wild felid [8] and listed as endangered under Taiwan’s Wildlife Conservation Act since 2009 [9]. Although once widespread throughout the island’s lowland regions, leopard cat populations currently exist in only three of Taiwan’s thirteen counties [10]. The leopard cat is considered a forest edge species which is predominantly found in unprotected, rural, human-dominated landscapes [10] where habitat fragmentation and degradation, as well as direct and indirect anthropogenic mortality (poisoning, illegal trapping, road kills, pesticides, domestic dog predation and disease transmission), threaten the species’ survival [11,12,13,14].

Within human-dominated landscapes, extinction risk is not only determined by the likelihood of exposure to anthropogenic threats but also by a species’ geographical, ecological and behavioral traits [15,16], such as the ability to adjust behavior in response to new challenges [17]. Carnivores which persist in areas with high human related risks have been shown to respond to anthropogenic threats through temporal and spatial behavioral adjustments, for example, by minimizing the risk of human encounters through fine-scaled habitat selection [18,19,20]. Despite its importance for conservation, quantitative information on the capacity and mechanisms for wildlife to co-exist with humans at fine spatial scales is scarce [21].

In an increasingly human-dominated landscape, it is vital to identify habitat features that influence fine-scaled habitat selection and facilitate co-existence [21,22,23]. Apart from suitable foraging habitats, nocturnal carnivores rely on resting habitats which allow them to spend the daytime hours when human activity peaks in safe refuge [18,22,24]. Resting habitats provides a place to rest, concealment from humans, and protection against predators and adverse weather conditions [22,24,25,26]. Consequently, the availability of resting habitats is positively related to species’ distribution and density [25,27,28].

Globally the leopard cat is widespread and classified as ‘Least Concern’ on the IUCN Red List [29]. The predominantly nocturnal [30] leopard cat persists in human-dominated and -modified landscapes [30], and human population density in itself does not seem to be a limiting factor for leopard cat occurrence [31,32,33]. Although some have suggested the species is tolerant to human presence [33,34], others have shown leopard cats avoid areas with high levels of human activity [11,31,32]. Possibly due to the difficulty of trapping and tracking this elusive species [35], published ecological studies on collared leopard cats are scarce and, with the exception of Grassman et al., 2005 (*n* = 20) [36], usually based on small sample sizes ([11] (*n* = 6); [32] (*n* = 1); [34] (*n* = 11); [37] (*n* = 4); [38] (*n* = 4, *n* = 7); [39] (*n* = 10)). Hence, little is known about the ecological needs of leopard cats at a microhabitat level, particularly in human-modified landscapes.

Conservation threats to leopard cats show high regional variability [30], it is therefore important to study the species at a regional scale to inform national conservation policy. In Taiwan, the first and only published ecological study on six radio-tracked leopard cats was conducted between 2006 and 2008 by Chen et al. (2016) [11], who studied home range sizes, movement and activity patterns in an area comprised of forest, grassland, agricultural land, and man-made constructions in the same county as our study. In this study, we radio-tracked seven leopard cats in rural Taiwan to determine how they utilize this human dominated landscape and investigate fine-scaled habitat selection for nighttime hunting and daytime resting. As suggested by Silmi et al. [34] and Rajaratnam et al. [39], we expected leopard cats to select agricultural habitats for nighttime hunting, and natural habitats with concealment as safe refuge for resting during the daytime when human and domestic dog activity peaks [11,35]. In accordance with other species [18,19,20,22], we expected leopard cats to avoid direct encounters with humans by selecting daytime resting sites further away from areas with high levels of human activity. We discuss how the results of this study can support Taiwan’s national leopard cat conservation strategy.

## 2. Materials and Methods

### 2.1. Study Area

This study took place in Zhuolan, a rural township in Miaoli County, situated in north-central Taiwan (Figure 1) at an elevation of 503 m. Zhuolan has relatively high and stable leopard cat numbers compared to other townships [7]. There are 6003 households in Zhuolan, with 2257 farmers who are generally positive about leopard cats and their conservation [40]. Roughly half of the households in this rural area own domestic dogs (49%), which are often kept as free-roaming (>36%) [40]. Main land use types in Zhuolan are orchards (predominantly pear, orange, and grape) and secondary forests [41] (Figure 1). A large river, the Da’an river, runs through the area. Mean monthly temperature was 24.7 ± 0.8 °C (mean ± SE), with a monthly rainfall of 162.5 ± 40.6 mm (mean ± SE). The area has a subtropical climate with hot/wet summers and colder/drier winters, most rainfall occurs from May–August and the highest temperatures are measured between June and September [42].

### 2.2. Trapping, Collaring and Tracking

Between May 2019 and May 2020, we trapped eight leopard cats: three females and two males residing along the Da’an river (river cats) and three males residing in the mountainous agricultural area of Zhuolan (mountain cats) (Figure 1) (Appendix A). The leopard cats were trapped using a 207.5-XL collapsible Tomahawk cage trap (Tomahawk Live Trap, Hazelhurst, WI, USA), with a custom-made separate bait cage, with two live coturnix quails, *(Coturnix japonica)* attached to its side [43]. Trapped leopard cats were immobilized by qualified veterinarians through remote injection of dexmedetomidine hydrochloride (25 μg/kg) and tiletamine hydrochloride/zolazepam hydrochloride (1.5 mg/kg), administered through the mesh of the trap via blowpipe (Sun-Yee Medical Technology Co., Taiwan). Immobilized leopard cats were sampled and collared with a 32–67 g VHF collar (Advanced Telemetry Systems, Isanti, MN, USA) with a nylon sewing elastic drop-off device [43]. The VHF collars transmitted a signal of 30 beeps/min, with a mortality signal of 45 beeps/min after 12 h without movement. The leopard cats were hand-injected with an antidote of 0.30 ± 0.03 mg/kg (mean ± SE) atipamezole hydrochloride and kept in the cage trap until fully recovered. For a detailed description of the method used for trapping and immobilization, see van der Meer et al. (2022) [35].

Post-release, we radio-tracked the leopard cats daily with a RA23-K VHF antenna and TR-4 (Telonics Inc., Mesa, AZ, USA) or Icom IC-R30 receiver (Icom Inc., Kamiminami, Osaka, Japan). Leopard cats were tracked between May 2019 and July 2021 for continuous periods of time (Appendix A). One female (MZF03) was killed by domestic dogs only four days after trapping (Appendix A). The remaining seven individuals were followed for 38–445 days (Appendix A). Three males died within the first three months of tracking: one was killed by domestic dogs, one by collision with a car, and one cause of death remains unknown. The remaining four individuals (two females and two males) were tracked for 421 ± 40 days (mean ± SE) (range 361–445 days) until their collars dropped-off and/or stopped working (Appendix A). The method of trapping and tracking of leopard cats was approved by the Institutional Animal Care and Use Committee of the National Pingtung University of Science and Technology (Permit no: 106003, 106014, 107041) and the Forestry Bureau (Permit no: 1061702029, 1080208595, 1090203226, 1091610844).

To ensure statistical independence of successive locations [39], consecutive daily locations of resting sites were always taken > 15 h apart, with an average of 24.00 ± 0.04 h (mean ± SE). We used existing infrastructure (e.g., small farm roads) to get as close to the leopard cats as possible and pinpointed their location based on triangulation from ≥three measurements [44,45] using the Neukadye field triangulation application (Neukadye, Golden, CO, USA) [43]. Resting sites were always located after 07:00 h at ca. 09:17 ± 00:02 h (mean ± SE). Occasionally (*n* = 4), an individual would still be active, in which case we returned later to locate its resting site.

Every ten days we tracked one of the collared leopard cats throughout the night. We started the tracking session when the individual became active and ended when the individual started resting. During these night sessions the leopard cat was located continuously at 30 min intervals, using triangulation based on ≥ three measurements taken within 7.32 ± 0.04 min (mean ± SE). On the rare occasions (*n* = 3) where, upon our arrival, the leopard cat was already active, we postponed the night tracking to the next night. In the morning, the leopard cat was considered to have stopped being active if it had been stationary at one location for ≥ 1 h.

Based on estimation with a hidden transmitter (*n* = 15), we obtained a triangulation accuracy of 8.3 ± 1.0 m (mean ± SE), at a distance of 59.3 ± 5.8 m (mean ± SE).

### 2.3. Land Use and Vegetation Assessment

We tracked the leopard cats for a total of 1876 days and 55 nights (Appendix A). Excluding the first three days after immobilization when leopard cat movement is atypical (*n* = 21) [35] and days when an individual went missing (*n* = 2) or was possibly already injured or dead (*n* = 6), this resulted in 1847 resting site locations and 1157 nighttime locations. Within each leopard cat’s home range boundary (95% minimum convex polygons (MCPs)) with stationary arithmetic mean), we generated random point locations by using the QGIS (QGIS Development Team, Open Source Geospatial Foundation, Beaverton, OR, USA) random points tool: 60 random points for individuals which were followed < 100 days, 120 for individuals which were followed ≥ 100 days (total *n* = 660). In some cases, resting sites (*n* = 106) or random points (*n* = 42) could not be taken into account in the analyses because the terrain did not allow us to physically visit the location or land use had suddenly changed (e.g., due to clearing or mudslides). With 5.0 ± 1.6% (mean ± SE) (range 0–10.9%) non-assessed locations per individual, this is unlikely to lead to bias.

At each nighttime location, daytime resting site, and random point, we determined land use classes and subclasses based on the classification of Taiwan’s National Land Surveying and Mapping Center [46]. Additionally, we also assessed habitat structure, vegetation and other ground cover at daytime resting sites and random points. Considering our mean triangulation accuracy, we, assisted by GPS and landmarks, visually estimated the percentages of vegetation and other ground cover types at ground level [47] within a 10 m radius from resting sites and random points. Following Edwards’ (1983) [48] definition of primary growth form types, vegetation was categorized as trees, shrubs, grasses, and herbs. Because of their dominance and interrelatedness to specific land use types [46], we recorded reed and bamboo separately. Ground cover types other than vegetation consisted of stones, bare ground, and manmade structures, such as buildings and roads. In addition, we classified habitat structure as open (visibility > 10 m), medium (visibility > 2 m < 10 m) or closed (visibility < 2 m) at leopard cat ground level. We acknowledge this provides a rough estimate of habitat composition.

### 2.4. Statistical Analysis

During the study period, both females had a litter of kittens (maternal period) (Appendix A). In accordance with Schmidt et al. (2003) [38], preliminary analyses using Mann–Whitney U tests showed this did not affect the duration of a night session, activity time, distance moved, or speed of movement (*p* ≥ 0.08). We therefore included both periods in our analysis of nighttime activity. Maternal dens are used for a consecutive period of time, and the selection of such dens is based on different criteria than selection of resting sites [26]. Maternal dens were therefore excluded from the home range and resting site analyses.

#### 2.4.1. Home Ranges

The reliability of home range estimates increases with the standardization of the sampling regime [44], and we therefore only included the daytime locations in our home range analyses. Home range area curves, size of home ranges, and core areas within those home ranges were calculated with the use of Biotas software (Ecological Software Solutions Inc., Sacramento, CA, USA). All home range area curves reached an asymptote for the number of sampling locations [49]. We used minimum convex polygons (MCPs) and, to take intensity of use into account, fixed kernels (FKs) [50] based on 95%, herewith eliminating occasional outliers that occur outside an individual’s “normal” range [51,52]. The size of the core areas was determined based on 50%MCPs and 50%FKs.

We performed a preliminary analysis using common algorithms for MCPs [51] and regular methods for FKs (least square cross validation, reference, and ad hoc smoothing parameters (h)) [52,53]. MCPs based on stationary arithmetic means (SAM) and FKs with smoothing factor h_ref_ provided the most reliable results. Although least square cross validation (LSCV) is the recommended method to calculate FKs [54], in accordance with field studies on other species, which show site fidelity and intensive use of core areas [50,55], LSCV resulted in underestimation and created discontinuous islands of utilization for our leopard cats. Small numbers of observations can cause the overestimation of FKs [54]; this seemed to be the case for at least one individual (MZM01). Overlap in home ranges was therefore calculated based on 95%- and 50%MCPs with stationary arithmetic means (SAM). Because home ranges may shift when adjacent home ranges become vacant due to mortality [56], we only took simultaneously tracked individuals into account in the calculations of home range overlap.

We used Taiwan’s National Land Surveying and Mapping Center’s land use investigation shapefile [57] and QGIS clipping and area calculation tools to calculate the percentages of land use types within 95%MCPs (SAM, excl. maternal period).

#### 2.4.2. Nighttime Activity and Daytime Resting Sites

For each night session (*n* = 55) we calculated the duration of a night session, activity time, distance moved, and speed with which leopard cats moved through the landscape. The duration of a night session is the time between the leopard cat’s first and last activity. Based on a maximum triangulation error of 14.3 m, activity was defined as movement ≥ 15 m. Activity time is the duration of a night session, minus the time an individual was inactive (movement < 15 m) during the night. Distance moved is the sum of distances between consecutive locations, while speed was calculated by dividing the distance moved by activity time.

We assessed land use selection patterns for nighttime activity and daytime resting based on a Jacobs’ index [58]. The Jacobs’ index follows the equation [59]:*D* = (*r* − *p*)/(*r* + *p* − 2*rp*),
where *r* is the proportion of habitat used and *p* the proportion of habitat available. *D* ranges between −1 (strong avoidance) and +1 (strong preference). Following Hayward et al. (2011) [60], we considered values between −0.2 and 0.2 as an indication that habitat is used as expected based on availability. The proportion of habitat used was based on the land use classes and subclasses at the point locations taken during the night sessions (*n* = 1157) or the daytime resting sites (*n* = 1551), while habitat available is based on land use types at the random points (*n* = 618). Additionally, for resting sites we also assessed selection patterns based on vegetation and other ground cover, and habitat structure at resting sites and random points. To avoid bias, we only included land use types for which total sample size of random points and/or resting sites was >5.

The land use shapefile provided insufficient detail for fine-scaled measurements, we therefore used Google Earth Pro [61] to measure elevation and the nearest distance of resting sites and random points to natural (forest or riverine) habitat, orchards, public tar roads, and buildings. We also measured the distance between resting sites on consecutive days and determined habitat patch size in which resting sites and random points were located, the number of times a patch was used, and the distance of the resting sites and random points to the patch’s edge. A patch was defined as an area with homogenous vegetation and elevation, not intersected by roads or rivers. All measurements in Google Earth Pro were taken by the same observer at an eye altitude of 600 m and corrected for changes in land use based on information collected on location in the field.

The variables elevation, patch size and distances of resting sites to patch edge, roads, rivers, buildings, orchards, and natural habitats did not follow the normality assumption. We therefore used Mann–Whitney U tests to test for differences between resting sites and random points. To avoid pseudo-replication [62], these tests were performed per individual leopard cat, based on which we determined whether a general pattern emerged, e.g., whether >50% of the leopard cats showed this selection pattern or whether there was a clear division between gender (males vs. females) and/or area (mountain vs. river). We used a one-tailed Spearman’s correlation to assess whether the number of times a patch was used (divided by the number of days a leopard cat was tracked to correct for possible bias due to the number of observations) was related to the ratio natural versus agricultural habitat within a home range (95%MCP SAM excl. maternal period). Statistical analyses were performed with SPSS software version 20.0 (SPSS Inc., Chicago, IL, USA).

## 3. Results

### 3.1. Home Ranges

Home range sizes varied considerably between individual leopard cats (Table 1; Figure 1). Based on 95%MCPs (SAM, excl. maternal period), there seemed to be both inter- and intrasexual variation in home range sizes (Table 1; Figure 1). For simultaneously tracked leopard cats, maximum home range overlap of 95%MCPs (SAM, excl. maternal period) was 17.7% and non-existent for 50%MCPs (SAM, excl. maternal period). However, coat pattern identification [63] of individual leopard cats in camera trap footage and roadkill records collected outside this study showed that 95%MCPs (SAM, excl. maternal period) of radio-tracked leopard cats overlapped with at least 2.4 ± 0.7 (mean ± SE) (range 1–5) non-collared individuals. Home range overlap was inter- as well as intra-sexual (Figure 1).

Agricultural land made up a substantial proportion (43.7–78.7%) of the land use types within 95%MCP (SAM, excl. maternal period) leopard cat home ranges (Table 1; Figure 1). The availability of natural habitat within a leopard cat home range varied considerably, with a maximum of 51.6% and a minimum of 8.0% (Table 1; Figure 1).

### 3.2. Nighttime Activity

Leopard cats started to be active around 17:57 ± 00:07 h (mean ± SE) (range start: 16:16–21:23 h) and ended their nighttime activity around 06:05 ± 00:09 h (mean ± SE) (range end: 01:20–08:43 h). Nighttime activity on agricultural land (orchards) took place between 19:09 ± 00:46 h and 06:03 ± 00:58 h (mean ± SE) (range start-end: 18:02–06:23 h). The duration of a night session was 12.33 ± 0.18 h (mean ± SE) (range 7.37–15.68 h). The total time leopard cats were active during the nighttime was 11.03 ± 0.24 h (mean ± SE) (range 6.67–14.00 h). The distance moved by leopard cats during the night was 2.51 ± 0.17 km (mean ± SE) (range 0.70–6.04 km), at a speed of 0.23 ± 0.01 km/h (mean ± SE) (range 0.06–0.48 km/h). With the exception of one night session, during which the individual MZF04 moved 200 m outside its usual range, nighttime activity fell within the leopard cat’s home range boundaries.

At night, leopard cats avoided manmade and, to a lesser extent, agricultural habitat or used it according to availability (Table 2). In general, leopard cats selected natural habitats for their nighttime activity (Table 2): river cats selected riverine habitats (Table 2), while mountain cats selected forest and, to a lesser extent, riverbeds (Table 2) (for a description of habitat types see Appendix A). The exception to this pattern was the female leopard cat MZF06, which selected unused land, abandoned orchards, and orchards.

A little over two-thirds (67.5%) of the point locations taken during the night sessions were within natural (forest or riverine) habitat: 28.3% on agricultural land and 4.2% on roads. Although, overall nighttime land use of mountain cats (70.6% natural, 28.0% agricultural) was comparable to river cats (66.1% natural, 28.4% agricultural), mountain cats less often spent an entire night session exclusively in natural habitats (5.9% mountain cats vs. 47.4% river cats).

A tar county road (road 140) notorious for leopard cat road kills [14], runs through MZM02 and MZF06′s home ranges (Figure 1). MZM02 and MZF06 crossed this road 42.9% and 54.5% of the night sessions, 1.33 ± 0.33 (mean ± SE) times and 3.00 ± 0.50 (mean ± SE) times per night, respectively. While MZF06 selected the road 140, MZM02 avoided it; nevertheless, it was this individual which was killed on this road by collision with a car. The other leopard cats either avoided the tar and dirt roads within their home ranges or used them according to availability (Table 2).

### 3.3. Daytime Resting Sites

Leopard cats’ situated resting sites in habitat patches equal to or larger (U ≥ 362.00, *p* ≤ 0.04) than the 11,463.66 ± 776.26 m^2^ (mean ± SE) patches in which random points (*n* = 618) were placed (Table 3). Compared to patches with random points, patches with resting sites were re-used more frequently (U ≥ 718.50, *p* < 0.01) (Table 3). The re-use of patches for resting was not related to the ratio natural versus agricultural habitat within a home range (R = −0.39, *p* = 0.19). Leopard cats generally rested at sites which were close to or inside natural habitat and frequently selected resting sites further away from orchards (U ≥ 523.00, *p* ≤ 0.01), buildings (U ≥ 1837.00, *p* ≤ 0.04), and, to a lesser extent, roads (U ≥ 1639.50, *p* < 0.01) than random points (Table 3). The only exception was the female MZF06, which rested at sites which were further away from the river (U = 14,631.00, *p* < 0.01) and closer to orchards (U = 14,794.00, *p* < 0.01), buildings (U = 14607.50, *p* < 0.01), and roads (U = 14,565.00, *p* < 0.01). Compared to random points, mountain males situated resting sites further from the edge of habitat patches (U ≥ 482.00, *p* ≤ 0.04).

Leopard cats selected closed natural vegetation to rest (Figure 2, Table 4). River cats selected riverine habitats and, to a lesser extent, unused land (Table 4) (Appendix A; Appendix A), while avoiding the riverbed (Table 4). Mountain cats selected forest, riverbeds, and, to a lesser extent, unused land and abandoned orchards (Table 4) (Appendix A–e; Appendix A). Abandoned orchards utilized for resting consisted of overgrown banana (35.0%), palm (27.5%), or fruit (45.5%) trees. Leopard cats avoided used orchards for resting (Table 4).

Leopard cats generally preferred resting sites with shrubs, reed, and stones, including the concrete river blocks used by Taiwan’s River Authority to manage flood water (Table 5). These tripod shaped blocks are usually overgrown with natural vegetation, herewith creating a semi-natural environment which provides safe refuge and shade (Appendix A). Trees were generally utilized according to availability, while bare ground, herbs, and manmade structures were mostly avoided for resting (Table 5). Resting sites on consecutive days were situated 1.21 ± 0.03 km (mean ± SE) apart for mountain males, 0.87 ± 0.05 km (mean ± SE) for river males and 0.78 ± 0.03 km (mean ± SE) for river females (Table 3).

### 3.4. Land Use Types Selected for Nighttime Activity vs. Daytime Resting

In our study, the leopard cat’s land use type selection pattern for nighttime activity showed many similarities with the species’ selection pattern for daytime resting (Table 6). However, there were also differences, e.g., the avoidance of areas with high levels of human activity (orchards and roads) was stronger during daytime resting (Table 6).

## 4. Discussion

Resting sites are especially important for nocturnal species in human-dominated landscapes because they need to spend the daytime hours, when human and domestic dog activity peaks [11,35], in safe refuge [24]. Within such human-dominated landscapes, some species utilize man-made structures for shelter, even when natural alternatives are available, while others solely rely on remaining patches of natural habitat [64]. Regardless, resting habitats contain features which provide protective cover from adverse weather conditions [25,26], predators [26], and humans [20,22,24]. Although it has been suggested that leopard cats utilize natural or semi-natural habitats for resting [34,39], little is known about the species’ fine-scaled selection of resting habitat. In our study, in rural Taiwan, leopard cats preferentially selected resting sites in natural closed (visibility < 2 m) vegetation with shrub, reed, and stones. Leopard cats in the mountainous agricultural area selected forest and riverbeds for resting (Appendix A; Appendix A), leopard cats residing along the Da’an river selected riverine habitat (Appendix A; Appendix A) but avoided the riverbed. This contrasting selection pattern for riverbeds results from riverbeds in the mountainous agricultural area usually being overgrown with reed, while the riverbed of the larger Da’an river is open and flanked by riverine habitat (Appendix A; Appendix A). In addition to natural habitat, leopard cats also selected semi-natural habitat with low levels of human activity for resting, i.e., unused land, overgrown abandoned orchards, and overgrown river blocks used to manage flood water (Appendix A). Based on our results, patches in which resting sites are situated should ideally be >1 ha with a radius of ≥13 m, situated ± 1 km apart.

Contrary to our expectations, leopard cats also selected natural habitats for their nighttime activity. Habitat selection during the nighttime when leopard cats hunt for prey is believed to be related to the availability of preferred prey species [34,37,39]. Due to ample food availability, prey can be substantially more abundant in agricultural habitats [39]. In our study area, agricultural habitat predominantly consists of fruit orchards [41]. There are no data available on prey density on these orchards; however, fruit farmers do not perceive rodents as a major threat to their crops [65]. In addition, camera trap data collected at our leopard cat trap locations at the interface of natural habitat and orchards in the farmland, and natural habitats along the Da’an river indicate a higher availability of murids and birds along the river (Appendix A). This higher prey availability correlates with higher leopard cat occurrence along the Da’an river [35]. Possibly due to the use of pesticides [14,65], prey density in orchards may thus be lower than expected. This could explain why the leopard cats in our study area preferred natural (forest or riverine) habitats for nighttime hunting, while man-made and agricultural habitats were avoided or used according to availability.

As shown for other wild felids [66,67] prey catchability rather than prey availability can determine habitat selection for nighttime hunting. Rajarathnam et al. (2007) [39] suggest that leopard cats prefer oil palm plants for hunting because the openness of these areas improves visibility and thus the catchability of prey. However, for stalking predators, there seems to be a fine balance between being able to find prey and having sufficient concealment for the stalk. Feral cats spend more time hunting at sites with structurally complex and dense cover, with vegetation height having a stronger influence on site visitation time than prey cues [68]. In accordance, lions prefer areas with good cover where prey is easier to catch over short-grass prey-rich areas [66]. While some suggest leopards preferably hunt in dense habitat [69], others found leopards select areas with medium cover where prey is easier to detect but cover is sufficient for stalking, rather than dense or open habitat [67]. The vegetation in the orchards in our study area is often kept short to facilitate farming activity and repel snakes (personal comm. farmers), due to lack of concealment, this may reduce prey catchability and provides an alternative explanation as to why leopard cats in our study area avoided agricultural habitats for nighttime hunting.

Speed of movement and mean distance moved by leopard cats during nighttime fell within the range reported by Chen et al. (2016) [11] and Schmidt et al. (2003) [38]. In accordance with other studies [11,30], the leopard cats in our study predominantly showed a nocturnal and crepuscular activity pattern, in which they were active for ca. 11 h. The timeframe during which the leopard cats were active on agricultural land reduced temporal overlap with humans and domestic dogs [11,35]. Despite being their preferred habitat, leopard cats in the mountainous agricultural area less often spent an entire night in natural habitat than leopard cats residing along the river. Together with the larger distance between consecutive daytime resting sites, this seems to indicate a higher level of habitat fragmentation in the mountainous agricultural area.

The underreporting of the exact method used to estimate home range sizes prevents accurate comparison across studies and highlights a need for standardization [70]; however, the sizes of our leopard cat home ranges and core areas, as well as overlap, seem to fall within the range reported by others [11,34,36,38,39]. Even though the proportion of natural habitat within leopard cat home ranges varied considerably, we found no relationship between the ratio of the proportion of natural versus agricultural habitat within a home range and the repeated use of natural habitat patches for resting. This suggests that the selection of resting sites is a fine-scaled process [18], with selection being based on habitat quality rather than quantity [22].

Although not as high as Chen et al., 2016 [11], who experienced a 100% mortality rate of radio-collared leopard cats within one year, the mortality of our leopard cats was considerable (Appendix A). In our study, confirmed causes of leopard cat mortality were vehicle collision and domestic dog attacks. With at least 50 road kills recorded between 2012 and 2017, vehicle collision is considered a main threat to leopard cat survival in Taiwan [12]. In addition, free-roaming domestic dogs have been shown to directly (predation) and indirectly (disease transmission) affect leopard cat survival [12]. Despite domestic dogs posing a threat to wildlife conservation [12,71], farmers frequently leave dogs to roam free [40], resulting in pack formation and high encounter rates [35]. In our study area, farmers’ attitude towards leopard cats is generally positive [40], which may be why, unlike in other parts of Taiwan [11], illegal trapping and poisoning were not found to be primary causes of leopard cat mortality in our study area.

Despite anthropogenic mortality being a main threat to their survival, leopard cats occur in habitats with human activity, such as logged forests and oil palm and sugar cane plantations [30]. It has therefore been suggested the species is tolerant to human presence [33,34]. However, although human population density in itself may not be a limiting factor for leopard cat occurrence [31,32,33], the species does seem to avoid direct encounters with humans, e.g., by steering clear of residential areas, man-made structures, and paved roads [11,31,32]. Comparable results were found in our study: areas with high levels of human activity were avoided, particularly during daytime resting. Resting sites were situated further away from areas with high levels of human activity (orchards, buildings, and roads) than random points. In addition, leopard cats residing in the mountainous agricultural area situated resting sites deeper inside natural or semi-natural habitat patches, thus avoiding contact with humans at the patch edge. This suggests that, similarly to the antipredator behavior of prey [72], leopard cats respond to variation in human derived risk by using a fine-scaled proactive (in response to a priori assessment of risk level) strategy to avoid direct encounters with humans. Similar results have been reported for brown bears *(Ursus arctos)*, wolves *(Canis lupus)*, and lynx *(Lynx lynx)* which, although inhabiting habitats with high levels of human activity, proactively avoid direct encounters with humans by adapting their resting site selection [18,20,22].

## 5. Conclusions

Leopard cats seem capable of surviving in human-dominated and -modified landscapes [30]; however, their long-term persistence in such landscapes will depend on balancing the needs of humans and wildlife. Initiatives to reduce direct and indirect anthropogenic mortality are beneficial to leopard cat conservation, but there is also a need to provide leopard cats with adequate foraging and resting habitats, which enable the species to avoid anthropogenic threats.

To maintain carrying capacity for leopard cats, it is important to take the availability and preservation of a mosaic of sufficiently large patches of (semi-)natural habitats for foraging and resting into account in regional land use planning and conservation strategies. Especially as there are several activities in rural Taiwan which may further reduce suitable leopard cat habitat, such as the destruction of riverine habitat through the harvest of river stones [73,74] and the conversion of natural habitat, abandoned orchards, and unused land into solar panel parks [75,76]. However, there are also opportunities to promote the conservation of leopard cat habitat, for example, through the leopard cat conservation performance payment scheme which was recently introduced in rural Taiwan [77]. Under such a scheme, farming communities could be encouraged to leave natural habitat at field edges, leave abandoned orchards or unused land undeveloped [78], or to set aside a percentage of land for wildlife conservation [79].

As for most other leopard cat studies [11,32,34,37,38,39], with seven collared individuals, the number of leopard cats followed in our study was small. This can impose statistical constraints and reduces the power to identify relationships among populations [80]. Nevertheless, we were able to detect consistent patterns in fine-scaled habitat selection by our studied leopard cats. We therefore believe that our study can assist in identifying and preserving suitable (semi-)natural habitats to support leopard cat conservation. However, it is important to note that leopard cat habitat use and conservation threats vary widely across study sites [30] and as such there is no one-fit-for-all solution and regional studies will be necessary to inform national leopard cat conservation policies.

## Figures and Tables

**Figure 1 animals-13-00234-f001:**
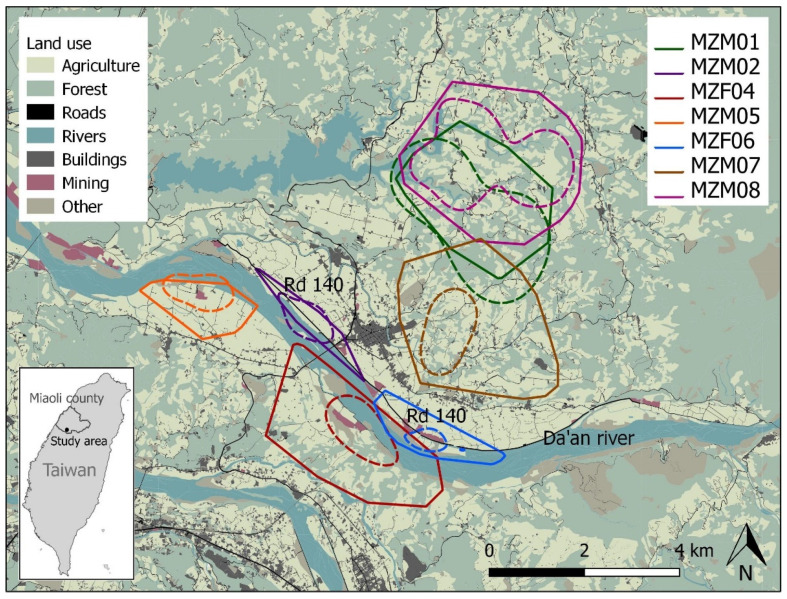
Map of the study area. Land use types within the study area and home range sizes (95% minimum convex polygons (MCPs) with a stationary arithmetic mean = solid line) and core area sizes (50% fixed kernels (FKs) with h_ref_ = dashed line) for our radio-tracked leopard cats (excluding the maternal period).

**Figure 2 animals-13-00234-f002:**
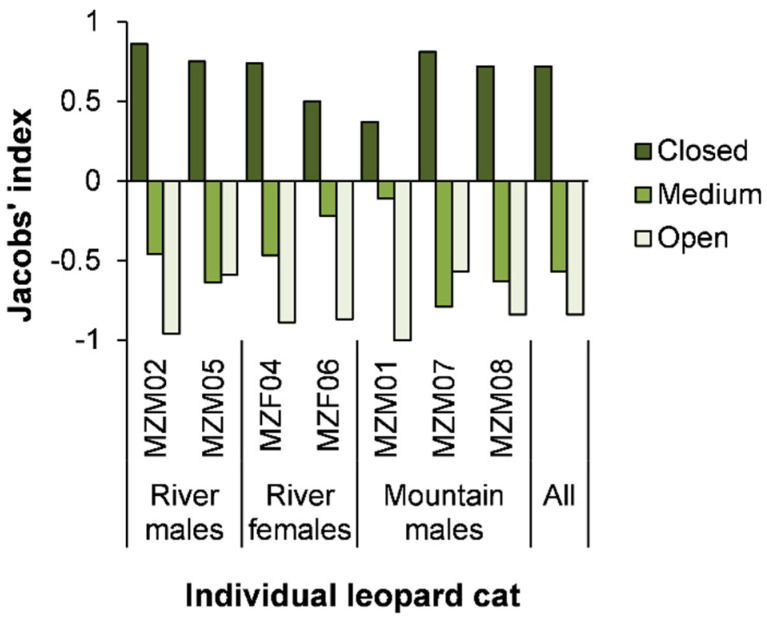
Jacobs’ index for vegetation structure at leopard cat daytime resting sites. Vegetation structures are closed, medium or open and selected (>0.2), avoided (<− 0.2), or used according to availability (−0.2–0.2).

**Table 1 animals-13-00234-t001:** Number of daytime locations on which home range calculations were based (excluding the maternal period for the river females), home range and core area sizes in km^2^ for individual leopard cats, based on 95% and 50% minimum convex polygons (MCP) with a stationary arithmetic mean (SAM), 95% and 50% fixed kernels (FK) with a reference smoothing parameter (h_ref_) and percentage of land use types within the 95%MCPs with SAM.

	River Males		River Females		Mountain Males			All	
Individual Leopard Cat	MZM02	MZM05	MZF04	MZF06	MZM01	MZM07	MZM08	Mean	SE
*n*	77	68	331	361	32	358	430		
Home range and core area size (km^2^)									
MCP95% SAM	0.97	1.88	6.83	1.86	6.68	8.45	9.67	5.19	1.34
FK95% h_ref_	3.94	4.94	9.85	2.70	24.79	12.58	14.66	10.49	2.93
MCP50% SAM	0.25	0.53	1.35	0.36	4.07	2.31	3.51	1.77	0.59
FK50% h_ref_	0.69	0.86	1.36	0.30	6.56	1.54	5.15	2.35	0.93
Land use types within MCP95% SAM (%)									
Agriculture	63.4	78.7	43.7	50.1	44.7	54.5	47.0	54.6	4.8
Natural	26.6	8.0	51.6	37.4	51.6	35.2	48.5	37.0	6.0
Manmade	9.6	9.5	4.6	11.6	3.8	8.7	4.1	7.4	1.2
Other	0.4	3.8	0.1	0.9	0	1.6	0.4	1.0	0.5

**Table 2 animals-13-00234-t002:** Jacobs’ index for land use classes (in bold) and subclasses at leopard cat nighttime locations: land use types are selected (>0.2), avoided (<−0.2), or used according to availability (−0.2–0.2), only land use types with a total sample size for nighttime locations and/or random points of >5 were included.

Land Use	River Males		River Females		Mountain Males			
Type	MZM02	MZM05	MZF04	MZF06	MZM01	MZM07	MZM08	All
**Agriculture**	**−0.06**	**−0.14**	**−0.54**	**0.26**	**−0.35**	**−0.28**	**−0.41**	**−0.24**
Abandoned	0.25	−1.00	−1.00	0.35	−1.00	0.20	−0.03	−0.11
Arable	−1.00	−1.00	−1.00	−	−	−1.00	−	−1.00
Orchard	0.04	−0.07	−0.35	0.28	−0.17	−0.22	−0.42	−0.16
**Natural**	**0.36**	**0.39**	**0.54**	**−0.23**	**0.44**	**0.36**	**0.50**	**0.33**
Forest	−	−	−0.74	−	0.31	0.32	0.56	−0.03
Riverbed	−1.00	0.69	−0.08	−0.64	0.75	1.00	−0.03	−0.16
Riverine	0.64	0.43	0.80	−0.16	−	−	−1.00	0.47
Unused	−0.19	−0.51	−0.40	0.87	−1.00	0.32	−0.32	−0.03
**Manmade**	**−0.57**	**−0.88**	**−0.20**	**−0.08**	**−1.00**	**−0.47**	**−0.84**	**−0.53**
Building	−1.00	−1.00	−	−1.00	−1.00	−1.00	−1.00	−1.00
River blocks	1.00	−	−	−	−	−	−	1.00
Dirt road	−1.00	−0.28	−	0.12	−	−	−	−0.30
Tar Road	−0.74	−1.00	−0.20	1.00	−1.00	0.20	−0.79	−0.36

**Table 3 animals-13-00234-t003:** Description of leopard cat daytime resting site characteristics: habitat patch size, distance to patch edge, number of times used, elevation and nearest distance to roads, rivers, buildings, and orchards for leopard cat resting sites, plus an overview of the significant outcomes of the Mann–Whitney U tests for differences between resting sites (Rest) and random points (Rnd) for the individual leopard cats.

					Number of Individuals with Significant (*p* < 0.05) Outcome Mann–Whitney U Test	
Variable	Mean	SE	Min	Max	Rest > Rnd	Rest < Rnd
Habitat patch size (m^2^)	12,101.19	363.96	106	164,952	3	0
Distance to patch edge (m)	12.94	0.30	1	133	3	0
No times patch used	5.88	0.14	1	24	6	0
Elevation (m)	390	1.55	257	590	0	2
Distance to road (m)	135.17	3.97	3	1731	3	1
Distance to river (m)	194.32	4.25	0.50	1056	1	3
Distance to building (m)	127.95	4.17	3	1702	4	1
Distance to orchard (m)	89.72	4.22	0	1715	5	1
Distance to natural habitat (m)	19.58	1.86	0	1126	0	7
Distance between consecutive resting sites (km)	1.00	0.02	0.002	4.71	-	-

**Table 4 animals-13-00234-t004:** Jacobs’ index for land use classes (in bold) and subclasses at leopard cat daytime resting sites: land use types are selected (>0.2), avoided (<−0.2), or used according to availability (−0.2–0.2), only land use types with a total sample size for resting sites and/or random points of >5 were included.

Land Use	River Males		River Females		Mountain Males			
Type	MZM02	MZM05	MZF04	MZF06	MZM01	MZM07	MZM08	All
**Agriculture**	**−0.22**	**−0.80**	**−0.88**	**−0.76**	**−0.52**	**−0.87**	**−0.72**	**−0.77**
Abandoned	0.80	−0.06	0.18	−0.34	−0.45	0.43	0.53	0.27
Arable	−0.46	−1.00	−1.00	-	-	−1.00	-	−0.93
Orchard	−0.83	−0.80	−0.87	−0.79	−0.48	−0.97	−0.91	−0.88
Pine	-	-	-	−0.51	-	1.00	1.00	0.47
**Natural**	**0.60**	**0.89**	**0.88**	**0.81**	**0.52**	**0.89**	**0.76**	**0.82**
Bamboo	-	-	−0.16	-	−1.00	−0.51	1.00	−0.12
Forest	-	-	−0.04	-	0.46	0.62	0.64	0.41
Riverbed	−0.84	−0.06	−0.82	−0.87	0.34	1.00	0.28	−0.58
Riverine	0.75	0.60	0.80	0.39	-	-	0.24	0.43
Unused	0.23	0.74	−0.09	0.94	0.34	0.74	−0.20	0.46
**Manmade**	**−1.00**	**−1.00**	**−1.00**	**−1.00**	**−0.21**	**−0.79**	**−0.87**	**−0.91**
Building	−1.00	−1.00	-	−1.00	−1.00	−1.00	−1.00	−1.00
Dirt road	−1.00	−1.00	-	−1.00	-	-	-	−1.00
Tar Road	−1.00	−1.00	−1.00	-	−1.00	−1.00	−1.00	−1.00

**Table 5 animals-13-00234-t005:** Jacobs’ index for vegetation and other ground cover types at leopard cat daytime resting sites: vegetation and other ground cover types were selected (>0.2), avoided (<−0.2) or used according to availability (−0.2–0.2), only vegetation and other ground cover types with a total sample size for resting sites and/or random points of >5 were included.

Type	River Males		River Females		Mountain Males			
	MZM02	MZM05	MZF04	MZF06	MZM01	MZM07	MZM08	All
Trees	0.28	−0.09	0.30	0.11	0.16	−0.05	−0.07	0.06
Shrub	0.30	0.79	0.30	0.65	−0.19	0.54	0.43	0.49
Reed	0.51	0.64	0.61	0.01	0.32	0.63	0.61	0.43
Grass	−0.19	−0.38	−0.14	0.79	−1.00	−0.20	0.00	0.22
Herbs	−0.10	−0.31	−0.43	−0.10	−0.21	−0.31	−0.38	−0.29
Bamboo	-	-	−0.30	-	0.02	0.30	0.12	0.10
Stones	0.84	0.67	−0.20	0.28	-	1.00	1.00	0.32
Bare	−0.66	−0.19	−0.66	−0.91	0.36	−0.37	−0.33	−0.58
River blocks	1.00	0.74	-	1.00	-	-	-	0.74
Building	−0.81	−1.00	-	−1.00	−1.00	−1.00	−1.00	−0.96
Dirt road	−1.00	−1.00	-	−1.00	-	-	-	−1.00
Tar road	−1.00	−1.00	−1.00	-	−1.00	−1.00	−1.00	−1.00

**Table 6 animals-13-00234-t006:** Description of land use types selected by leopard cats for nighttime activity versus daytime resting, percentages between brackets represent the percentage of studied leopard cats which showed this selection pattern for the land use types available within their home ranges.

Land Use Type	Nighttime Activity	Daytime Resting
Agriculture	Orchards were usually avoided (42.9%) or used according to availability (42.9%). Where available, arable land was avoided (100%). Abandoned farms were avoided (42.9%), used according to availability (28.6%) or selected (28.6%).	Orchards and, where available, arable land were avoided (both 100%). Abandoned farms were selected (42.9%), used according to availability (28.6%) or avoided (28.6%).
Natural	Mountain cats selected forests (100%), while river cats selected riverine habitat (100%). Riverbeds were frequently selected by mountain cats (66.7%) but avoided by river cats (50.0%). Regardless of location, unused land was often avoided (57.1%).	Mountain cats selected forests (100%) and riverbeds (100%). River cats selected riverine habitat (100%) and mostly avoided the riverbed (85.7%). Regardless of location, unused land was frequently selected (71.4%).
Manmade	Buildings were avoided (100%), tar roads, and, where available, dirt roads were mostly avoided (57.1% and 66.7% respectively)	Buildings, tar roads, and, where available, dirt roads were avoided (all 100%)

## Data Availability

The authors confirm that the data supporting the findings of this study are available within the article and its Appendix A. Raw data supporting the conclusions of this article are available from the authors upon reasonable request.

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
