# Peer review of "Fine-Scaled Selection of Resting and Hunting Habitat by Leopard Cats (Prionailurus bengalensis) in a Rural Human-Dominated Landscape in Taiwan"

_animals, 2023, doi:10.3390/ani13020234_

Round 1
Reviewer 1 Report
This is very interesting work on a species of conservation importance. The work is thorough and explicit in its attempts to define the types of habitats used as resting spots for leopard cats. Overall the paper is very well written, see individual comments below.
Simple Summary and Abstract has some grammar issues. Not as many with introduction
Line 13: Nowadays is a bit too casual of a term
Line 60: Would be good to list IUCN status
Lines 79-82: Nice description of resting habitat and its importance
Line 100: Should list names in addition to adding reference numbers.
This is a great map (Fig. 1) but telling the difference between the land use categories with the different shades of grey is very difficult.
Line 117: No need to say Fig 1 again.
Lines 118 and 121: Will need to add reference for website, only one at end of paragraph is necessary
Line 133: Somewhere in methods you should list the permit number and authorizing agency for these captures.
Line 145: Same, add reference for website
Line 149: Watch grammar here—“got killed” is not proper.
Line 151: I would be more specific on the time tracked here. If less than a full year, it comes into question if enough time was allotted for seasonal variation in behavior, hormone cycles, etc.
Line 172-redundant to lines 151-152. Better here with more accurate tracking days
Line 197: Section 2.4 should also tell the types of statistical tests performed during your analysis and computer programs used.
Line 208-209: Reference 47, need more info here on how this was achieved…”made sure” requires specific elicitation.
Line 211: MCP is used before and the paper and only defined here. Should be defined on first use (even if in figures or tables)
Line 215: The first 2 sentences in this paragraph are quite run-on. Consider dividing.
Line 256: This should be a numbered reference
Line 260: Throughout you put a dash behind resting and not sure as to why
Table 1: The lines corresponding with Individual leopard cat (that are sideways) are difficult to read and some are cut off. These should likely not be sideways as it is confusing which belongs to which column. Looks much better the way it is laid out in Table 2
Line 321: I think you mean collision
Figure 2. I cannot tell the difference between these gray shades, recommend using color or hatch marks. Figure caption has extra line space.
Line 375: U would start this paragraph with “the leopard cats in our study…..” Start strong with what your study found first, then acknowledge past research
Discussion: The first few sentences of the paragraph reads like a reiteration of the results. You should point out why each of the findings were significant and the implications for the findings as you do in the rest of the discussion.
Line 418: This paragraph and the next seem like they should be earlier in discussion as this is the point of the paper (perhaps even the lead). Also need to address the small sample size in this study and its relations to results.
Line 462: No need for (semi-), just semi-natural.
Line 495: A little too casual
Overall, really great manuscript!
Author Response
Reviewer #1 This is very interesting work on a species of conservation importance. The work is thorough and explicit in its attempts to define the types of habitats used as resting spots for leopard cats. Overall the paper is very well written, see individual comments below. Thank you for taking the time and effort to review our manuscript. We sincerely appreciate your valuable comments and suggestions, which helped us in improving the manuscript. Simple Summary and Abstract has some grammar issues. Not as many with introduction Line 13: Nowadays is a bit too casual of a term Based on this comment, we changed nowadays into presently, see L13. Line 60: Would be good to list IUCN status We added this information to L82-83. Lines 79-82: Nice description of resting habitat and its importance Thank you. Line 100: Should list names in addition to adding reference numbers. Thanks for pointing this out, we added the names, see L101. This is a great map (Fig. 1) but telling the difference between the land use categories with the different shades of grey is very difficult. Thanks, based on your suggestion and the suggestions from the other reviewer, we decided to change this black and white figure into a color figure. The difference between the various land use types should now be more clear. Line 117: No need to say Fig 1 again. Fig. 1 was removed from this sentence, see L117. Lines 118 and 121: Will need to add reference for website, only one at end of paragraph is necessary The website was removed, a reference with the website details was added to L120. See also reference 42 in the reference list. Line 133: Somewhere in methods you should list the permit number and authorizing agency for these captures. We added this information to the method section, see L152-155. Please note this information was also provided in the Institutional Review Board Statement L544-547. Line 145: Same, add reference for website In the first paragraph of 2.2 Trapping, Collaring and Tracking we used company details instead of websites. To be consistent, we therefore decided to add the company details between brackets where we refer to products (L144-145, L160-161 and L181) and added references where we refer to data sets (L120, L234 and L261). Line 149: Watch grammar here—“got killed” is not proper. We changed ‘got killed’ into ‘was killed’. See L148. Line 151: I would be more specific on the time tracked here. If less than a full year, it comes into question if enough time was allotted for seasonal variation in behavior, hormone cycles, etc. We added more detail to this sentence, see L150-151. A detailed overview of the number of days each individual was tracked can also be found in Table S1. We appreciate your comment about the time allotted for seasonal variation in behavior, hormone cycles etc. The focus of our paper is fine-scaled selection of hunting and resting habitat. With the exception of maternal sites, for the two females and two males we have been able to track for > 361 days, preliminary analyses showed no indication of seasonal variation in selection of hunting and resting habitat nor did we find an indication of seasonal variation in home range sizes. We therefore felt confident we could include hunting and resting locations of the three individuals which were tracked for < 81 days in our analysis. Line 172-redundant to lines 151-152. Better here with more accurate tracking days This sentence was meant to give a general indication of sample size. More detailed information about tracking days was provided in L150-151, see comment above. Line 197: Section 2.4 should also tell the types of statistical tests performed during your analysis and computer programs used. We are not entirely sure if we have interpreted this suggestion correctly. Based on this comment, we added the type of statistical test used to L204-206. Details of the statistical tests and computer programs used to analyze Home Ranges and Nighttime Activity and Daytime Resting Sites can be found under sub header 2.4.1 and 2.4.2 respectively. Line 208-209: Reference 47, need more info here on how this was achieved…”made sure” requires specific elicitation. Your comment made us realize we did not correctly phrase this sentence, the use of the words ‘made sure’ is incorrect, you can only check if an asymptote is reached. We used biotas to check if the home range area curves reached an asymptote for the number of sampling locations. We rephrased this section to make this clear, see L213-216. Line 211: MCP is used before and the paper and only defined here. Should be defined on first use (even if in figures or tables) Thanks for pointing this out, we made changes accordingly to the figure caption of Fig. 1 and L179. Line 215: The first 2 sentences in this paragraph are quite run-on. Consider dividing. We followed your suggestion and, to improve readability divided this sentence in two, see L220-223. Line 256: This should be a numbered reference A reference was added here, see L261 and reference 60 in the reference list. Line 260: Throughout you put a dash behind resting and not sure as to why Changed accordingly throughout the manuscript. Table 1: The lines corresponding with Individual leopard cat (that are sideways) are difficult to read and some are cut off. These should likely not be sideways as it is confusing which belongs to which column. Looks much better the way it is laid out in Table 2 Yes, we agree this Table was impossible to read and changed the lay-out into the lay-out used for Table 2. Line 321: I think you mean collision Yes, we definitely meant collision, thanks for pointing this out. Collusion was changed into collision, see L334. Figure 2. I cannot tell the difference between these gray shades, recommend using color or hatch marks. Figure caption has extra line space. We followed your suggestion and changed this figure into a color figure. We hope this has improved the clarity of this figure. Line 375: U would start this paragraph with “the leopard cats in our study…..” Start strong with what your study found first, then acknowledge past research Thank you for this and the suggestions below, based on which we restructured the discussion. We now start the discussion with resting site habitat selection, followed by nighttime habitat selection. The more general results like distance moved and speed of movement during nighttime, as well as home range sizes, are presented shortly in a paragraph halfway the discussion L447-464. In this paragraph we have also added the information related to the percentage of natural habitat available within a home range L459-464, which was originally part of the conclusion. This way the discussion stays more focused on the main findings of the study. Discussion: The first few sentences of the paragraph reads like a reiteration of the results. You should point out why each of the findings were significant and the implications for the findings as you do in the rest of the discussion. Based on your suggestion, we have restructured the discussion, see explanation above. Line 418: This paragraph and the next seem like they should be earlier in discussion as this is the point of the paper (perhaps even the lead). Also need to address the small sample size in this study and its relations to results. Based on your suggestion, we have restructured the discussion, see explanation above. The small sample size was addressed at the end of the conclusion in the paragraph in which we also mention the regional variability of leopard cat habitat use. See L514-518. Line 462: No need for (semi-), just semi-natural. Changed accordingly, because we intended to mention both natural (in some cases there still remains small patches of natural habitat within the agricultural areas) and semi-natural habitat we included ‘natural and..’ to this line as well, see L487-488. Line 495: A little too casual Based on this comment we changed ‘it has to be kept in mind’ into ‘it is important to note that’, see L520. Overall, really great manuscript! Thank you, we appreciate the compliment.
Reviewer 2 Report
Line 19 – should be “30-minute”; same for line 164
L43 – change to “12.9 ± 0.3 m”
L44- you state: “Reduced availability of resting sites is linked to declines in population density and distribution”, but I don’t see your data documenting either decline. You might say “...resting sites likely is linked....” unless you have data.
L52 – not all “remaining” wildlife habitat is “embedded within a human-modified matrix” unless you define the entire planet as a human-modified matrix (i.e., there are vast areas of unmodified habitats, some protected, some not protected). Change the wording to reflect this.
L83 – reference “nocturnal leopard cat” – Ross et al 2015?
L97 – change to “and activity patterns in an area comprised of forests, grasslands, agricultural land, and man-made construction and in the same county as our study.”
L100 – delete “by”
Fig. 1 – I think I see a difference between agriculture areas and “white” areas. Are the white areas developed areas? Regardless, label them.
L117, 118 – change to “24.7 ± 0.8 °C” and “162.5 ± 40.6”
L142,143 – change to “For a detailed description.....immobilization, see van der Meer et al. (2022) [34].”
L170 – change to “8.3 ± 1.0” and “59.3 ± 5.8” – you can’t measure these values to 10-cm accuracy.
L182 – if 10.9% is the maximum, give a mean value and the range of values.
L189 – change to “10-m”
L203-204 – did you test to see if it differed? Since these sites are even more important to population persistence that daily rest sites, give these data as a separate section in this paper, regardless of the small sample size (who else has such data?)
L207 – you should still present nightime home range sizes, as long as you describe your methods and sample sizes, and then assess how different they are from “daytime” range calculations. You had to pick areas by which to calculate use and availabilities, but using daytime locations to assess nighttime movements and locations seems at odds, unless you reassure the reader that its “all the same”.
L245 – “where” should be left justified and not capitalized (with a comma after the equation above).
L255 – “,” should be “;”
L270 – “... based on which we determined whether a general pattern emerged.” – describe this, perhaps by giving actual examples for where you identified a pattern and for where you did not.
L284 = “study, showed” should be “study showed”
L285 – should be “2.4 + 0.7” – no need to talk about hundredths of an animal
L296-304 – Make a figure showing actual amount of activity in each hourly interval during the night.
L309,310 – should be “The exception to this pattern was female leopard cat MZF06, which selected unused land, abandoned orchards and orchards.”
L311 – “two-thirds”
L315 – instead of “...were less often able to spend an entire night...” do you mean “...less often spent an entire night...”?
L312 – should be “collision”
L389 – present (and cite at the end of this first sentence in this paragraph) a simple table that summarize the differences between night-time/active and daytime/resting habitat selection – this is a key finding that needs to be emphasized.
L442 – how did you calculate a mortality rate of 45%? Not counting MZF03, you had 3 mortalities in 1,876 monitoring days, giving an annual survival rate of 56% (or mortality of 44%!)
Author Response
Reviewer #2
Thank you for taking the time and effort to review our manuscript. We sincerely appreciate your valuable comments and suggestions, which helped us in improving the manuscript.
Line 19 – should be “30-minute”; same for line 164
Changed accordingly, see L19 and L167.
L43 – change to “12.9 ± 0.3 m”
Changed accordingly, see L43
L44- you state: “Reduced availability of resting sites is linked to declines in population density and distribution”, but I don’t see your data documenting either decline. You might say “...resting sites likely is linked....” unless you have data.
Thanks for pointing this out. This sentence was indeed based on references, not on the data from our study. We considered changing this sentence but because it is embedded in between two sentences which directly relate to our study felt this may not solve the problem entirely. We therefore decided to delete this sentence from the Abstract.
L52 – not all “remaining” wildlife habitat is “embedded within a human-modified matrix” unless you define the entire planet as a human-modified matrix (i.e., there are vast areas of unmodified habitats, some protected, some not protected). Change the wording to reflect this.
We tried to emphasize we are referring to small patches of remaining habitat and changed the wording of this sentence into ‘Small patches of remaining wildlife habitat are often embedded within a human-modified matrix’. See L51-52.
L83 – reference “nocturnal leopard cat” – Ross et al 2015?
This reference is indeed Ross et al 2015 and was added to L83.
L97 – change to “and activity patterns in an area comprised of forests, grasslands, agricultural land, and man-made construction and in the same county as our study.”
As per your suggestion, this addition was made to L97-98.
L100 – delete “by”
Instead of deleting ‘by’ we, as per the suggestion of the other reviewer, added the full names of the references. See L101-102.
Fig. 1 – I think I see a difference between agriculture areas and “white” areas. Are the white areas developed areas? Regardless, label them.
Based on your suggestion and the suggestions from the other reviewer, we decided to change this black and white figure into a color figure. The difference between the various land use types should now be more clear.
L117, 118 – change to “24.7 ± 0.8 °C” and “162.5 ± 40.6”
Changed accordingly, see L117-118.
L142,143 – change to “For a detailed description.....immobilization, see van der Meer et al. (2022) [34].”
Changed accordingly, see L141-142.
L170 – change to “8.3 ± 1.0” and “59.3 ± 5.8” – you can’t measure these values to 10-cm accuracy.
Thank you for pointing this out, changed accordingly, see L173.
L182 – if 10.9% is the maximum, give a mean value and the range of values.
As per this suggestion, we added mean ± SE and the range of values to L186-187.
L189 – change to “10-m”
Changed accordingly, see L193.
L203-204 – did you test to see if it differed? Since these sites are even more important to population persistence that daily rest sites, give these data as a separate section in this paper, regardless of the small sample size (who else has such data?)
We appreciate your suggestion and have considered this. However, given the uniqueness of these data, we have decided to present the results on characteristics of maternal sites in a separate manuscript with an explicit focus on breeding behavior.
L207 – you should still present nighttime home range sizes, as long as you describe your methods and sample sizes, and then assess how different they are from “daytime” range calculations. You had to pick areas by which to calculate use and availabilities, but using daytime locations to assess nighttime movements and locations seems at odds, unless you reassure the reader that its “all the same”.
We have considered this suggestion but decided against calculation of home ranges based on nighttime data for the following reason: Traditional home range estimators such as minimum convex polygons and kernel density estimators assume independence and identically distributed data. Nighttime locations, taken at 30-minute intervals, are highly clustered dependent locations. This means there is a high level of autocorrelation within these data sets, as a result of these locations being sampled closely in time and space (see also Silva et al. (2021) Auto-correlation informed home range estimation: a review and practical guide. Methods in Ecology and Evolution, doi: 10.1111/2041-210X.13786). To reliably calculate home ranges from nighttime locations it would therefore be best to take one point per night session into account, for example the center point. However, this would leave us with a maximum of 15 points per individual, which is insufficient to calculate MCPs and FKs.
Just to clarify, we did not use daytime locations to determine selection patterns for nighttime activity, we used random locations generated within each individual’s home range. To avoid confusion about the method used we changed the wording ‘random site’ into ‘random point’ throughout the document. In addition, we changed resting sites into days followed in L185 of the methods section.
With the exception of one night session of female MZF04 (which during that night moved 200 m outside her home range boundary in riverine vegetation comparable to the riverine vegetation within her home range), all night session fell within the leopard cat’s daytime location based home ranges. We are therefore confident that the method used provides a reliable indication of nighttime habitat selection patterns. To explain this to the readers, we added L311-313 to the results section.
L245 – “where” should be left justified and not capitalized (with a comma after the equation above).
Changed accordingly, see L250.
L255 – “,” should be “;”
Sorry, we are not entirely sure which part of L254 you are referring to and kindly ask you to clarify this suggestion.
L270 – “... based on which we determined whether a general pattern emerged.” – describe this, perhaps by giving actual examples for where you identified a pattern and for where you did not.
To clarify this sentence, we added examples of what we considered ‘general patterns’ to L274-276. The actual patterns are described in the methods section 3.3. Resting Sites, see also Table 3.
L284 = “study, showed” should be “study showed”
Changed accordingly, see L290.
L285 – should be “2.4 + 0.7” – no need to talk about hundredths of an animal
Changed accordingly, see L291.
L296-304 – Make a figure showing actual amount of activity in each hourly interval during the night.
Although we appreciate this suggestion, we decided not to include a figure of the amount of activity in each hourly interval during the night because we prefer the manuscript to remain focused on fine-scaled selection of nighttime hunting and daytime resting habitat.
L309,310 – should be “The exception to this pattern was female leopard cat MZF06, which selected unused land, abandoned orchards and orchards.”
Thank you for this suggestion, changed accordingly, see L318.
L311 – “two-thirds”
Changed accordingly, see L324.
L315 – instead of “...were less often able to spend an entire night...” do you mean “...less often spent an entire night...”?
Thanks for pointing this out, the phrase ‘were less often able to spent an entire night’ was based on the assumption that leopard cats would prefer to spend an entire night in natural habitat. The results section is not the right place for these kinds of assumptions. We therefore rephrased this sentence as per your suggestion, see L328 and added a possible explanation to the discussion, see L452-455.
L312 – should be “collision”
Thanks for noticing this mistake, collusion was changed into collision, see L334.
L389 – present (and cite at the end of this first sentence in this paragraph) a simple table that summarize the differences between night-time/active and daytime/resting habitat selection – this is a key finding that needs to be emphasized.
Based on this suggestion we have added Table 6 to the results section. Because we felt squeezing this table in under section 3.2. Nighttime Activity or 3.3. Daytime Resting Sites would not do it justice and would disrupt the flow of these sections, we decided to add a short section to the results: 3.4. Land use types selected for Nighttime Activity versus Daytime Resting. See L388-395.
L442 – how did you calculate a mortality rate of 45%? Not counting MZF03, you had 3 mortalities in 1,876 monitoring days, giving an annual survival rate of 56% (or mortality of 44%!)
Thank you for pointing out this mistake. Your comment made us realize that we did take MZF03 into account in the next sentence in which we describe mortality causes. To address this inconsistency, we decided to refer to Table S1 in the first sentence (rather than the percentage). We also rephrased this sentence to emphasize that the mortality rate in our study was not as high as in the study by Chen et al. 2016 see L465-467.
